# VividFace: A Robost and High-Fidelity Video Face Swapping Framework

Hao Shao[1]    Shulun Wang[2]    Yang Zhou[2]    Guanglu Song[2]
Dailan He[1]    Zhuofan Zong[1]    Shuo Qin[2]    Yu Liu[2] ✉    Hongsheng Li[1,3] ✉
[1]CUHK MMLab    [2]SenseTime Research    [3]CPII under InnoHK

## Abstract

Video face swapping has seen increasing adoption in diverse applications, yet existing methods primarily trained on static images struggle to address temporal consistency and complex real-world scenarios. To overcome these limitations, we propose the first video face swapping framework, VividFace, a robust and high-fidelity diffusion-based framework. VividFace employs a novel hybrid training strategy that leverages abundant static image data alongside temporal video sequences, enabling it to effectively model temporal coherence and identity consistency in videos. Central to our approach is a carefully designed diffusion model integrated with a specialized VAE, capable of processing image-video hybrid data efficiently. To further enhance identity and pose disentanglement, we introduce and release the Attribute-Identity Disentanglement Triplet (AIDT) dataset, comprising a large-scale collection of triplets where each set contains three face images—two sharing the same pose and two sharing the same identity. Augmented comprehensively with occlusion scenarios, AIDT significantly boosts the robustness of VividFace against occlusions. Moreover, we incorporate advanced 3D reconstruction techniques as conditioning inputs to address significant pose variations effectively. Extensive experiments demonstrate that VividFace achieves state-of-the-art performance in identity preservation, temporal consistency, and visual realism, surpassing existing methods while requiring fewer inference steps. Our framework notably mitigates common challenges such as temporal flickering, identity loss, and sensitivity to occlusions and pose variations. The AIDT dataset, source code, and pre-trained weights will be released to support future research. The code and pretrained weights are available on the project page.

## 1    Introduction

In recent years, face swapping has emerged as a crucial technology across various domains, from content creation [36] and privacy protection [52] to safe stunt scene production [39] and digital twin generation [35]. As video is a predominant medium for communication, the demand for high-quality face swapping techniques has grown substantially. Video face swapping involves extracting identity features from a source face and seamlessly integrating them with the attributes (such as expressions, poses, *etc.*) and background of a target face while maintaining temporal consistency. However, despite the recent advancements, current face-swapping methods encounter difficulties in video contexts, as most are optimized for static images rather than dynamic video sequences.

Existing face swapping approaches can be broadly categorized into three main methodologies: 3D-based, GAN-based, and diffusion-based methods. Traditional 3D-based methods [3, 4, 47, 34], primarily utilizing 3D Morphable Models (3DMM) [5], often struggle with low-resolution outputs and face blending issues. GAN-based approaches [33, 9, 27, 2, 29, 56] encounter challenges with training instability, mode collapse, and producing low-resolution output, particularly in complex

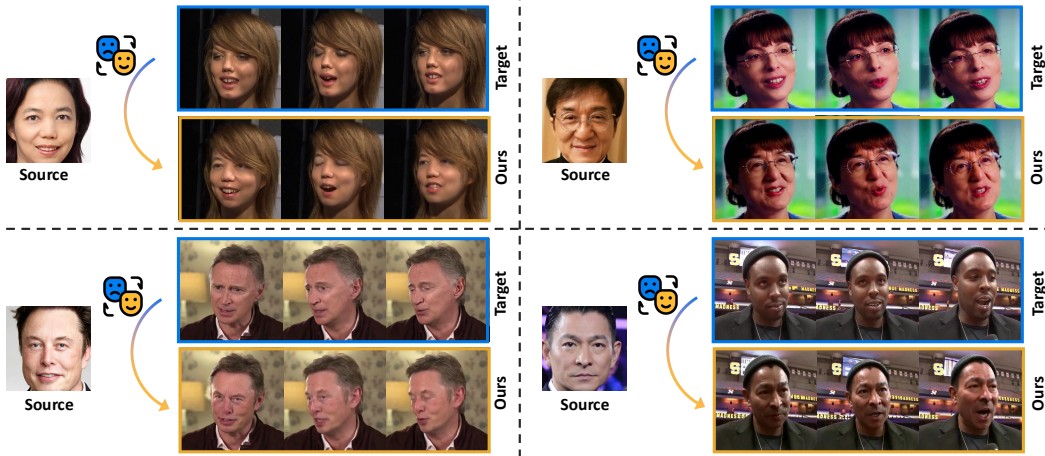

Figure 1: Face swapping results of VividFace at $512 \times 512$ resolution. Our method produces high-fidelity and vivid outputs that accurately follow both pose and expression changes. Corresponding videos are provided in the supplementary material.

cases. Recently, diffusion models [20] have gained prominence in image synthesis tasks, offering advantages such as high-fidelity output, enhanced controllability, and high training stability.

Recent advancements like DiffSwap [55] and REFace [1] have highlighted the effectiveness of diffusion models for image-level face swapping. However, significant challenges persist when extending these methods to video face swapping, including maintaining temporal consistency, handling large pose variations, and addressing occlusions. To overcome these challenges, we propose VividFace, the first robust and high-fidelity diffusion-based video face swapping framework. VividFace introduces an innovative hybrid training strategy that integrates diverse static image data with temporal video sequences, effectively overcoming the inherent limitations of using video-only data, such as insufficient diversity resulting from highly similar frames within individual videos. This hybrid approach significantly expands training data diversity, enhancing the robustness and generalization capability of our model. Our framework employs a specifically designed diffusion model optimized for processing both static images and temporal video data. We further introduce the VidFaceVAE adapted to jointly handle face images and video sequences, effectively mitigating temporal flickering typically encountered in existing video swapping methods.

To further enhance identity and attribute preservation, we create and release the Attribute-Identity Disentanglement Triplet (AIDT) dataset, consisting of 1 million image triplets and 0.6 million video triplets. Each triplet includes a source face, a target face with shared identity but different poses and expressions, and a GAN-generated face matching the target's pose and expression but featuring a distinct identity. This structured dataset significantly boosts the model's capability for disentangling identity from pose and expression. While ReliableSwap [53] also utilizes a triplet-based concept, it mainly addresses artifact reduction through augmentation rather than robust disentanglement. Additionally, we develop a comprehensive occlusion augmentation strategy that dynamically introduces various occluding objects over target faces, significantly improving the framework's robustness against real-world occlusions. To effectively manage significant pose variations, we incorporate a 3D Morphable Model (3DMM)-based reconstruction as additional conditioning input. This 3D guidance ensures accurate pose and expression representation, facilitating better generalization across diverse video contexts. To further reduce information leakage and enhance robustness, we retain only the pose and expression features from the reconstruction, discarding texture and identity information.

Experimental results demonstrate our framework's superiority in terms of Fréchet Video Distance (FVD), temporal consistency, and attribute/identity preservation, with fewer inference steps compared to existing methods. Besides, we also demonstrate the stability and generation of our method in multiple complex cases.

To summarize, this paper makes the following contributions:

- The first diffusion-based video face swapping framework, VividFace, featuring a novel image-video hybrid training strategy.

- We provide and plan to release a large-scale AIDT dataset to significantly improve face feature identity-expression disentanglement.
- Robustness enhancements through comprehensive occlusion augmentation and advanced 3D face reconstruction conditioning to handle large pose variations effectively.
- Extensive experimental analyses demonstrating superior temporal consistency, identity preservation, and visual quality, along with comprehensive ablation studies.

## 2 Related Work

### 2.1 Face Swapping

The frameworks of face swapping are generally categorized into three types: 3D-based [3, 4, 47, 34], GAN-based [33, 9, 27, 2, 29, 56, 26, 38], and diffusion-based methods [23, 55, 1, 19]. In addition to these three main approaches, FaceShifter [27] introduces a two-stage framework that generates high-fidelity swapped faces by thoroughly and adaptively exploiting and integrating the target attributes. 3D-based frameworks typically employ the parameterized 3DMM [5] model to reconstruct the swapped face. Face2Face [47] transferred expressions from source to target face by fitting a 3DMM face model to both faces. The authors in [34] show that face swapping with robust segmentation preserves identity in intra-subject swaps and reduces recognizability in inter-subject cases. HifiFace [51] introduced a semantic facial fusion module to improve photorealism. However, these 3D-based methods yield low similarity and unrealistic textures due to limited resolution.

GAN [16] has been a powerful tool for generating realistic synthetic images. The popular algorithm DeepFakes [11] utilizes an encoder-decoder architecture for identity-specific face swapping but lacks generalization. To improve adaptability, FSGAN [33] proposes a subject-agnostic approach with a recurrent reenactment module, inpainting and a blending module. E4S [29] reframes face swapping as fine-grained editing by disentangling shape and texture, using regional GAN inversion for precise feature manipulation and occlusion handling. SimSwap [9] introduces an ID Injection Module and Weak Feature Matching Loss for flexible, high-fidelity identity swapping. However, GAN-based methods often struggle with balancing losses and handling shape variations or occlusions, leading to inconsistencies in illumination and identity in complex cases.

Recently, diffusion models have become a leading framework for image & video generation. DiffFace [23] first leverages conditional diffusion models for stable identity-preserving swapping. DiffSwap [55] and FaceAdapter [19] build upon conditional inpainting paradigms to achieve high-fidelity, controllable swapping. REFace [1] improves this by reframing swapping as a self-supervised inpainting task. However, existing diffusion-based methods mainly target static images, overlooking key video challenges like temporal consistency, occlusions, and large pose variations.

### 2.2 Diffusion Models

Diffusion models [20, 31, 57, 58, 43] have recently emerged as a powerful generative framework, achieving state-of-the-art performance in various domains, including image synthesis [14, 20, 45], editing [22, 44, 28], super-resolution [50, 15], and video generation [18, 6, 32]. Unlike GANs, which often suffer training instability, diffusion models offer a more stable training process by gradually denoising data from random noise, resulting in high-fidelity outputs. Notable advancements include Stable Diffusion [40], which enhances efficiency by operating in the latent space, and SVD [6], which incorporates temporal modules to scale diffusion models for video tasks. Conditioning mechanisms like cross-attention and concatenation enhance controllability, enabling targeted generation across applications [48]. Thus, diffusion models are increasingly popular for versatile, high-quality content creation.

## 3 Method

### 3.1 Preliminaries

Our method employs Stable Diffusion (SD) [40] as the backbone network. Stable Diffusion is a text-to-image model built on the Latent Diffusion Model (LDM), which enables efficient image generation by operating within a compressed latent space. SD uses a variational autoencoder (VAE) [24] to map

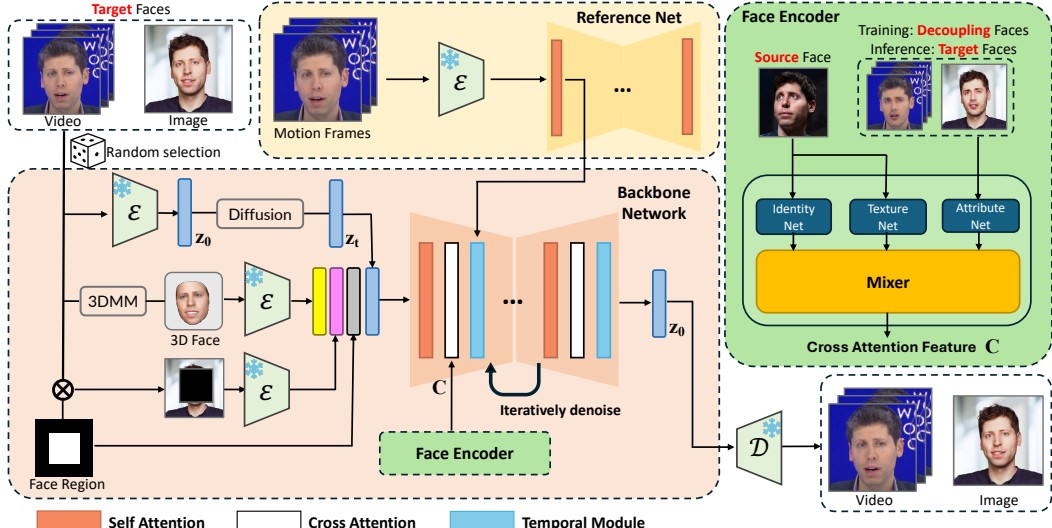

Figure 2: Overview of the proposed framework. During training, our framework randomly chooses static images or video sequences as the training data. In addition to the noise $z_t$, three other types of inputs are integrated to guide the generation process: (1) a face region mask, which controls the generation of facial imagery; (2) a 3D reconstructed face, which helps guide the pose and expression, especially in cases of large pose variations; and (3) masked source images, which supply background information. These inputs are processed through the Backbone Network, which performs the denoising operation. Within the Backbone Network, we employ cross-attention and temporal attention mechanisms. The temporal attention module ensures temporal continuity and consistency across frames. Our face encoder extracts identity and texture features from the target face, as well as pose and expression details from the source face, and uses these features to produce realistic and high-fidelity results.

the original image $x_0$ unto a latent representation $z_0$. reducing computational cost while preserving visual quality. The image is encoded as $z_0 = \mathcal{E}(x_0)$ and decoded back as $x_0 = \mathcal{D}(z_0)$. SD follows the Denoising Diffusion Probabilistic Model (DDPM) [20] framework, introducing Gaussian noise $\epsilon$ to the latent $z_0$ across timesteps $t$, generating a noisy latent $z_t$ over a series of steps. During inference, the model denoises $z_t$ back to $z_0$, guided by condition features. The denoising backbone $\epsilon_\theta$, based on a U-Net [41], is trained to predict the noise and remove it progressively, using the objective:

$$ L = \mathbb{E}_{t,c,z_t,\epsilon} \left[ \| \epsilon - \epsilon_\theta(z_t, t, c) \|^2 \right], $$

where $c$ represents text features derived from a CLIP encoder [37]. SD uses a U-Net with cross-attention mechanisms, to fuse text embeddings with latent features, enabling fine control over generated images based on text prompts. This allows SD to generate detailed, high-fidelity images while responding effectively to user input.

### 3.2 Hybrid Face Swapping Framework

**Video Face Swapping Task.** Video face swapping aims to seamlessly transfer a source face identity onto a target video while preserving the target's pose, expression, lighting, and background. Although recent works like DiffSwap [55, 23, 1] have demonstrated promising results for static image face swapping, extending these approaches directly to videos presents substantial challenges. These include temporal distortions, flickering, occlusion sensitivity, and difficulties in managing significant pose variations.

To effectively tackle these challenges, we propose VividFace, a diffusion-based video face swapping framework specifically designed for robust, temporally coherent, and high-fidelity results. VividFace introduces a novel hybrid training strategy that leverages abundant image-level data alongside temporal video data. This hybrid approach enhances diversity and robustness in training, significantly mitigating issues prevalent in video-only methods. Our framework initially encodes both source

images $x^i_{src} \in \mathbb{R}^{1 \times 3 \times H \times W}$ and video sequences $x^v_{src} \in \mathbb{R}^{T \times 3 \times H \times W}$ into a unified latent space $z_0 \in \mathbb{R}^{T \times C \times H \times W}$ using a specially designed VAE. Static images are treated equivalently as single-frame videos, ensuring consistent embedding. Subsequently, we train a conditional diffusion model $\epsilon_\theta(z_t, t; \mathbf{C})$ that performs latent space denoising, emphasizing temporal consistency and identity fidelity. Here, $\mathbf{C}$ represents the conditioning vectors, and $t$ indicates the denoising timestep. Due to the absence of ground truth data when source and target images originate from different individuals, our model uses pairs of face images from the same identity during training. As depicted in Figure 2, training batches alternate between static image data and video sequences, ensuring efficient gradient synchronization and optimal learning dynamics.

**VidFaceVAE.** As shown in Figure 3, our proposed VidFaceVAE is a VAE framework designed to enhance the reconstruction quality of facial data, effectively handling both video sequences and static images. The VidFaceVAE primarily consists of (2+1)D blocks, combining 2D spatial and 1D temporal convolutions to form pseudo-3D operators. For image inputs, the STFM (Spatial Temporal Fusion Module) outputs the result of the 2D ResBlock directly, bypassing the temporal ResBlock. For video inputs, the STFM combines the outputs from both the 2D and temporal blocks using a learnable coefficient $\beta$, described as $o = \beta \times o_{\text{spatial}} + (1 - \beta) \times o_{\text{temporal}}$, where $o_{spatial}$ and $o_{\text{temporal}}$ denote the output from the spatial branch and the temporal branch. We do not involve the temporal downsampling modules in our VAE framework as it needs to process image data. The (2+1)D structure of VidFaceVAE

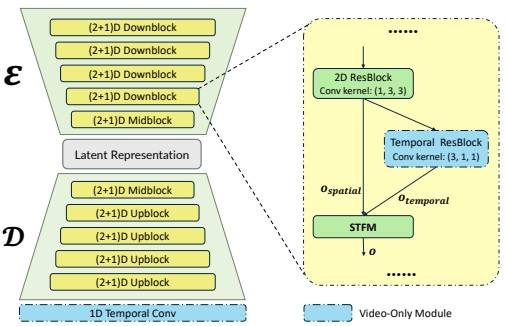

Figure 3: Overview of the proposed VidFace-VAE, capable of simultaneous encoding and decoding of both image and video data. Specific modules are designed for video inputs, which image inputs bypass when necessary.

provides two primary advantages: (1) it significantly reduces computational complexity compared to full 3D convolutions, and (2) it facilitates leveraging pretrained 2D VAE parameters and stable diffusion (SD) weights, accelerating training convergence and enhancing overall performance. In contrast to OD-VAE [8], our approach does not utilize 3D causal convolutions due to their limited capacity, particularly when processing static images, and our backbone architecture does not rely on transformer-based designs.

**Temporal Modules.** Inspired by recent approaches like EMO [48] and AnimatedDiff [18], we introduce self-attention temporal layers specifically designed for video sequences. During training with video data, we prepare additional motion frames $x^{motion}_{src}$ to enrich temporal context. The temporal attention mechanism combines these motion features with target frames at matching resolutions along the temporal dimension, improving temporal coherence. To smooth the first video clip's generation, motion frames are initialized as zero vectors during training.

### 3.3 Designs of Condition Vectors

In our framework, several carefully designed condition vectors are used to guide the generation process, ensuring accurate and consistent visual outputs for both static images and video sequences. We formulate video face swapping as a conditional inpainting task, where masked videos with cropped face regions provide the background and lighting conditions. The corresponding face regions guide the diffusion model on where are generated the faces.

In many in-the-wild videos, faces often exhibit significant pose variations, which can lead diffusion models to produce suboptimal results, such as facial distortions and inaccurate pose estimations. To address this issue, we propose using a 3D reconstruction technique to reconstruct the face and use its output as local guidance for pose and expression details. Specifically, we employ 3DMM [5] to extract BFM (Basel Face Model) coefficients, setting the texture and identity component to zero to reduce information leakage. Replacing the reconstructed face with the original target would introduce further leakage, as the ground truth face is identical to the input, which could impair the model's generalization ability. To ensure that the generated face maintains the same identity as the source

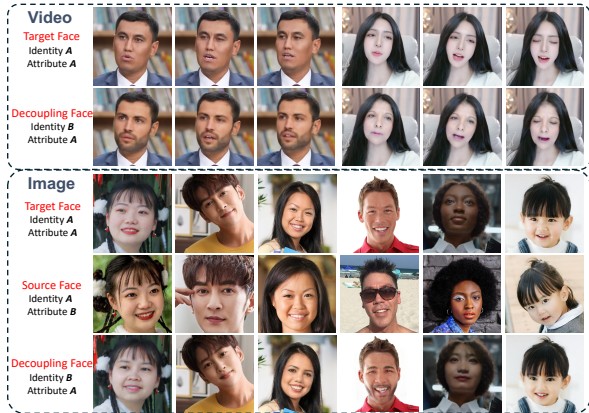

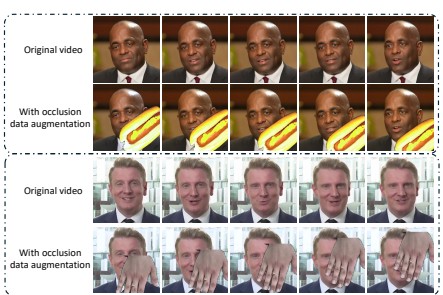

Figure 4: Visualization of our AIDT dataset, which includes 1M image triplets and 0.6M video pairs. For video facial data, we only present the target and decoupling faces, as source faces can be derived from any other frame within the same clip.

Figure 5: Visualization of our occlusion data augmentation, which improves the stability and consistency of the generated videos.

face while preserving attributes (such as pose, expression, *etc.*,), we inject cross-attention features **C** extracted by our face encoder as global context to the diffusion model.

**Face Encoder.** The face encoder module in our framework plays a critical role in extracting and integrating features from the target and source faces to guide the face-swapping process effectively. As illustrated in the right part of Figure 2, the face encoder is composed of three primary networks, each responsible for capturing distinct aspects of facial information: (1) identity net: this network focuses on extracting the core identity features from the target face; (2) texture net: this network is designed to capture detailed texture information from the target face, such as skin tone, fine facial features; (3) attribute net: the net extracts additional facial attributes from the source, such as pose, expression, and other dynamic features that contribute to a realistic and expressive representation.

The straightforward approach is to send the source image to both the identity and texture networks, while the target image is sent to the attribute network. However, a challenge arises when the source and target faces do not belong to the same person, as the ground truth is unavailable in the real world. In most previous methods [1, 55, 23], the source and target images are assumed to be the same, meaning all three networks receive identical input. This results in difficulties for the face encoder in extracting distinct features and leading to information leakage. Specifically, this leakage causes the model to merely "copy and paste" the face region, effectively completing the task by superficially transferring facial features without meaningful feature disentangling or transformation. In contrast, our framework, built on the AIDT dataset (shown in Figure 4), employs source images (same identity, but different attributes) and decoupling images (same attribute, but different identity). These images guide the face encoder to disentangle and fuse facial features, improving generalization across different identities during inference.

Within the Mixer module, the extracted features first undergo cross-attention operations to capture mutual dependencies. The outputs are then scaled by learned weights and fused via weighted sum. This process combines the identity, texture, and attribute features to create a comprehensive cross-attention feature representation **C**. This fused representation offers rich context to guide the diffusion model during face generation, ensuring high fidelity and identity consistency across video frames.

## 3.4 Training Strategy

Our training process involves three stages to progressively enhance model performance for video face swapping. The first stage focuses on training the VidFaceVAE , where we apply reconstruction, perceptual, and KL divergence losses to ensure high-quality reconstruction and a well-structured latent space. The training data primarily consists of facial images and videos. Given the specifically designed architecture, the spatial modules are initialized using the original 2D VAE. In subsequent stages, the VAE is frozen and no longer updated. In the second stage, we pretrain the model using image data, while the ReferenceNet and temporal modules of the backbone network remain inactive.

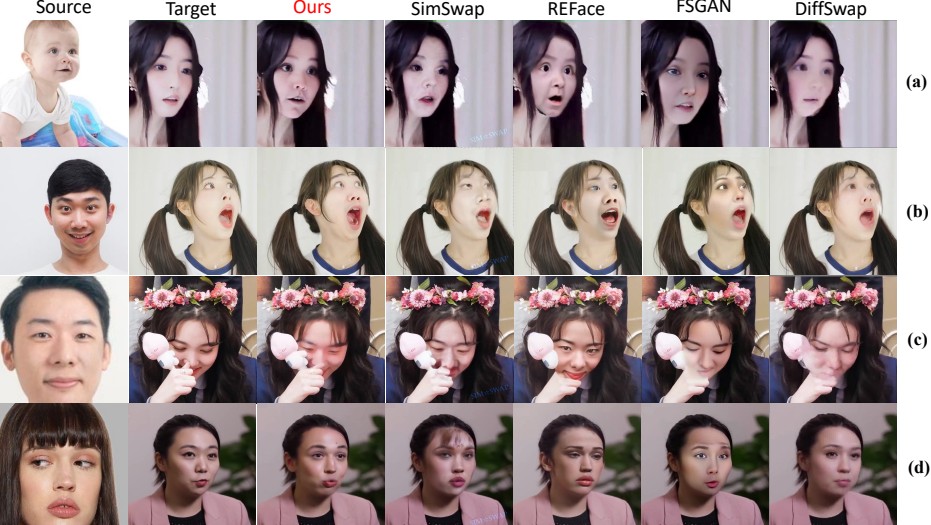

Figure 6: Qualitative comparison at $512 \times 512$ resolution. Our method generates high-fidelity results and handles challenging cases effectively, such as large poses (b) and occlusions (c).

The backbone is initialized from the original SD weights. Finally, we perform image-video hybrid training by activating temporal modules and introducing video data, initializing temporal modules from AnimateDiff [18] for effective temporal consistency and smooth frame transitions.

## 4 AIDT Dataset

In this section, we describe the construction of triplet pairs for our AIDT (Attribute-Identity Disentanglement Triplet) dataset, which includes 1M image triplets and 0.6M video triplets, as shown in Figure 4.The data was generated via a pipeline of collection, detection, tracking, and post-processing. See Appendix B for details. The dataset helps the face encoder to disentangle and fuse distinct facial components—ID features, texture features from the source face, and attribute features from the decoupling face. This enhances generalization, especially when the source and target faces belong to different individuals during inference.

In addition, Figure 5 showcases our occlusion augmentation pipeline. We collect hand and everyday object images, along with their corresponding masks, from web data sources. To simulate temporal dynamics, we design a motion trajectory animation scheme that incorporates scale, rotation, and translation over time. This augmentation strategy improves robustness under occlusion and enhances the temporal consistency of synthesized videos.

## 5 Experiment

Detailed information regarding dataset preparation, network architectures, hyperparameters, and training procedures can be found in Appendix A.

### 5.1 Evaluation Protocol

Considering that most previous baselines, such as CelebA [25] and FFHQ [21], are primarily focused on image face swapping, we propose a new benchmark for video face swapping, VidSwapBench. Our benchmark includes 200 source images and 200 high-resolution target videos, with each video containing 128 frames and a single trackable face. These videos and images feature unseen identities and backgrounds, ensuring a diverse and challenging dataset. To evaluate performance, we generate 200 swapped videos using our framework. For comparison, since other methods are based on image-level face swapping, we perform face swapping frame by frame for those methods. For facial data reconstruction, we use SSIM, PSNR and LPIPS [54] to evaluate the quality of reconstructed images and videos. For video face swapping, we use FVD [49] to assess the overall quality of the generated videos. The attribute transfer error is measured by pose and expression errors. We use HopeNet [42]

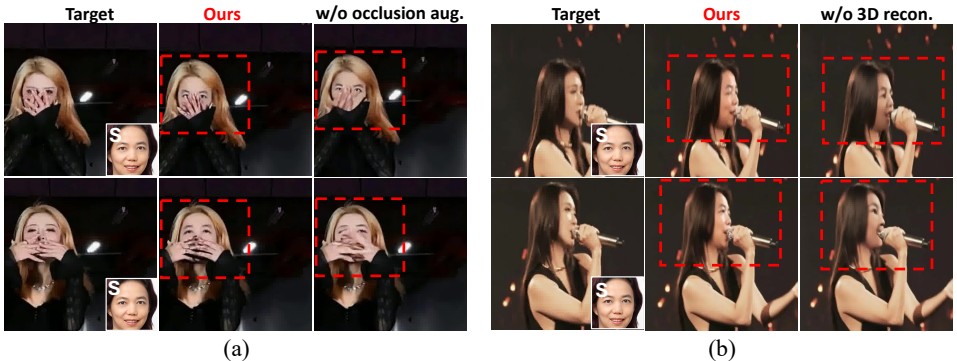

Figure 7: Ablation on the occlusion data augmentation and 3D face reconstruction.

and Deep3DFaceRecon [13] to detect these attributes, and the L2 distance to the ground truth is used as the evaluation metric. For ID retrieval, we extract identity features from the source images using ArcFace, and for each swapped video, we perform face retrieval by searching for the most similar faces among all source images. The retrieval is measured by the average cosine similarity of all frames, and we report the Top-1 and Top-5 accuracy.

Table 1: Comparison on VidSwapBench dataset.

| Method | $FVD_{32}\downarrow$ | $FVD_{128}\downarrow$ | ID@1↑ | ID@5↑ | Pose↓ | Expr.↓ |
|---|---|---|---|---|---|---|
| MegaFS [56] | 1280.3 | 200.7 | 72.8 | 82.1 | 6.21 | 0.74 |
| HifiFace [51] | 1377.3 | 386.4 | 74.1 | 82.9 | 6.10 | 0.74 |
| SimSwap [9] | 1242.8 | 186.6 | 76.5 | 88.5 | **5.12** | 0.76 |
| FSGAN [33] | 1507.9 | 423.8 | 24.5 | 40.0 | 5.19 | 0.73 |
| DiffFace [23] | 2404.7 | 1404.9 | 1.5 | 4.1 | 18.3 | 1.58 |
| DiffSwap [55] | 1530.2 | 809.3 | 14.5 | 26.3 | 12.9 | 1.02 |
| REFace [1] | 1336.9 | 311.9 | 71.9 | 86.5 | 6.67 | 0.91 |
| **Ours** | **1201.1** | **122.6** | **78.3** | **90.2** | 5.43 | **0.72** |

Table 2: Comparison on FFHQ dataset.

| Method | FID↓ | ID@1↑ | ID@5↑ | Pose↓ | Expr.↓ |
|---|---|---|---|---|---|
| MegaFS [56] | 12.0 | 59.6% | 74.1% | 3.33 | 1.11 |
| HifiFace [51] | 11.58 | 75.3% | 87.1% | 3.28 | 1.41 |
| SimSwap [9] | 13.8 | 90.6% | 96.4% | 2.98 | 1.07 |
| E4S [29] | 12.38 | 70.2% | 82.73% | 4.50 | 1.31 |
| DiffFace [23] | 8.59 | 87.2% | 94.4% | 3.80 | 2.28 |
| DiffSwap [55] | 8.58 | 78.2% | 93.6% | **2.92** | 1.10 |
| REFace [1] | 5.53 | 95.4% | 98.7% | 3.74 | **1.04** |
| **Ours** | **4.05** | **96.9%** | **99.2%** | 3.70 | 1.07 |

## 5.2 Comparisons with Existing Methods

**Qualitative Results.** Since videos cannot be displayed in the PDF and due to submission policy restrictions on showing generated videos, we provide several comparison videos in the supplementary materials and strongly encourage the reader to view them. We perform quantitative comparison at $512 \times 512$ resolution. As shown in Figure 6 (a) and (d), our method generates high-fidelity swapped faces, with attributes that closely match the target faces. In Figure 6 (b), our method successfully transfers both face shape and expression under large pose variations, benefiting from the 3D reconstruction mask, while other methods exhibit generation artifacts. In Figure 6 (c), where a toy and hand occlude the girl's face, most other methods fail to handle the occlusion properly, with the toy and hand either displaced or fused together. Additionally, many methods result in noticeable facial deformations. In contrast, our method successfully recovers the occluded areas and maintains accurate face swapping, thanks to our augmentation strategy.

**Quantitative Results.** In Table 1, we compare seven open-source methods (four GAN-based and three diffusion-based). The results show that our method outperforms others in ID retrieval and FVD, generating high-fidelity swapped face videos while preserving the source identity. It also achieves comparable performance in pose and expression, maintaining target attributes effectively. Furthermore, since our model supports both image and video face swapping, we also evaluate it on the standard FFHQ dataset. As shown in Table 2, VividFace achieves state-of-the-art results in FID and ID retrieval, while delivering comparable performance in pose and expression preservation.

## 5.3 Ablation Studies

We conducted comprehensive ablation experiments to analyze the contributions of different components and design choices within our framework. The quantitative results are summarized in Table 3, and qualitative visualizations are presented in Figure 7. **Hybrid vs. Static Training (Exp. 1):** training exclusively with static images results in decreased identity preservation (Top-1 accuracy

Table 3: Ablation on training strategies and module designs

| Exp Id | Method | $FVD_{32}\downarrow$ | $FVD_{128}\downarrow$ | ID retrieval↑ Top-1 | Top-5 | Pose↓ | Expr.↓ |
|---|---|---|---|---|---|---|---|
| 0 | Baseline | 1201.1 | 122.6 | 78.3 | 90.2 | 5.43 | 0.72 |
| 1 | Static Training | 1231.9 | 128.1 | 75.7 | 88.1 | 5.60 | 0.74 |
| 2 | Without 3DMM Guidance | 1197.4 | 121.3 | 78.1 | 90.4 | 5.55 | 0.74 |
| 3 | Merged ID and Texture Net | 1203.8 | 121.9 | 76.4 | 87.1 | 5.49 | 0.73 |
| 4 | Init. Texture/Attr Net with CLIP | 1237.9 | 138.2 | 76.0 | 86.1 | 5.30 | 0.72 |

Table 4: Comparison of different VAE architectures

| Architecture | | Facial videos | | |
|---|---|---|---|---|
| Encoder | Decoder | SSIM↑ | PSNR↑ | LPIPS↓ |
| 2D | 2D | 0.967 | 37.61 | 0.048 |
| 2D | (2+1)D | 0.976 | 38.77 | 0.039 |
| (2+1)D | (2+1)D | **0.983** | **41.11** | **0.027** |

reduced from 78.3 to 75.7) and higher FVD scores, underscoring the critical advantage of our hybrid training approach in achieving superior temporal consistency and identity fidelity. **Impact of 3D Reconstruction Conditioning (Exp. 2):** removing the 3DMM conditioning slightly reduces pose and expression accuracy, highlighting the effectiveness of 3DMM guidance in handling complex pose variations. **Separate Identity and Texture Networks (Exp. 3):** combining identity and texture features into a single network leads to reduced identity retrieval accuracy, emphasizing the necessity of separate networks for effectively disentangling and extracting distinct facial features. **Initialization Strategy (Exp. 4):** initializing texture and attribute networks with CLIP weights negatively affects identity preservation, demonstrating the effectiveness of our proposed initialization strategy. **Impact of Occlusion Augmentation:** the qualitative results in Figure 7 (a) show severe distortions when occlusion augmentation is omitted. Introducing occlusion augmentation substantially improves stability, consistency, and visual quality under occlusion scenarios. **Effect of 3DMM Conditioning on Large Pose Variations:** Figure 7 (b) illustrates that excluding 3D face reconstruction guidance results in significant instability and distortion for large pose variations.

**VAE Architecture Analysis.** Table 4 compares reconstruction performances of different VAE architectures. The baseline (SD-VAE) employs a pure 2D structure, the second model utilizes a (2+1)D decoder combined with a 2D encoder, and our proposed VidFaceVAE employs a full (2+1)D encoder-decoder. VidFaceVAE outperforms alternative architectures across all metrics, achieving the highest SSIM (0.983), PSNR (41.11), and lowest LPIPS (0.027). This clearly demonstrates the advantage of effectively integrating spatial and temporal processing in facial video reconstruction.

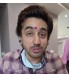
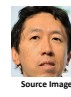

Figure 8: Ablation on the different combinations of texture weights and attribute weights.

## 5.4 Face Feature Mixing Analysis

We further examine the impact of varying texture and attribute weights within the face encoder. Figure 8 shows that increasing texture weight improves identity similarity but can degrade pose and expression preservation if too high. Higher attribute weights maintain target features but reduce identity fidelity. Optimal performance is achieved by balancing these weights, set to 1.0 for identity and 0.6 for texture and attribute in our experiments.

## 6 Conclusion

In this paper, we introduced a novel diffusion-based framework for video face swapping, addressing key challenges such as temporal consistency, identity preservation, and large pose variations. Our image-video hybrid training strategy leverages both static images and video data, improving model diversity and robustness. The VidFaceVAE , coupled with a custom Attribute-Identity Disentanglement Triplet (AIDT) dataset and 3D Morphable Model integration, enables accurate face swapping while mitigating issues like flickering and occlusions. Our framework outperforms existing methods in FVD, temporal consistency, and identity preservation, with fewer inference steps. Together with the released dataset, it provides a more efficient solution for high-quality video face swapping and lays the foundation for future advancements.

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

# A Implementation Details

We collected approximately 550 hours of facial videos from the internet to train our models, and the facial images are partially sourced from VGGFace2-HQ [10]. In our experiments, we use a latent space of size $13 \times 64 \times 64$ and a U-Net architecture for the $\epsilon_0$ denoising network. Images and video clips sampled from the dataset are resized and cropped to $512 \times 512$. The number of motion frames, $M$, is set to 4, and the generated video length, $T$, is set to 8 frames. For the face encoder, the identity network is based on ArcFace [12], while the texture and attribute networks are based on DINO [7]. We use the SCRFD [17] for facial bounding box detection. The mixing coefficients of the ace encoder are set to 1.0 for identity features, and 0.6 for both texture and attribute features. The experiments are conducted using 16 NVIDIA A100 GPUs and optimized with AdamW [30]. In the first stage of the VAE training, the learning rate is set to 5e-6 with a batch size of 32. The weights of reconstruction, perceptual, and KL divergence loss are 1.0, 0.1, 1e-6 respectively. For the second and third stages, the learning rate is increased to 1e-5, with the batch size remaining at 32. During inference, we generate video clips using the DDIM sampling algorithm for 32 steps.

# B AIDT Dataset details

This section describes our approach to constructing the AIDT (Attribute-Identity Disentanglement Triplet) dataset, which consists of two parts: video clip collection and triplet pairs construction.

## B.1 Video Clip Collection Pipeline

**Video Collection.** We collect facial videos from public platforms to capture a diverse range of visual and auditory content. The video corpus includes two primary categories: static content, such as news broadcasts, interviews, and public speaking events; and dynamic content, encompassing genres like travel guides, vlogs, and musical performances. This variety ensures a comprehensive dataset, representing both controlled and spontaneous human expressions and activities across different scenarios.

**Face Detection and Tracking.** For each video, we first apply a face detection model to identify face bounding boxes in each frame. Subsequently, all face detections are fed into a tracking procedure to generate face tracklets across frames. Each resulting tracklet is then split into video clips, each containing between 30 and 200 frames. Finally, we crop each video clip to center the head within the frame and to maintain a consistent aspect ratio.

**Data Post-Processing.** Based on the preliminary clips generated in the previous stage, we further refine the clips by applying three key constraints to collect final training data. We begin by utilizing HyperIQA [46] to filter out low-quality clips, setting an average quality assessment threshold score of 50 to ensure visual fidelity. Second, we enforce identity consistency within each clip by using a face recognition model to extract facial features and compute cosine similarity scores between every pair of frames, thus verifying that each clip represents a single individual. Finally, we utilize an OCR model to exclude clips containing text near the facial region, minimizing visual distractions that could interfere with model training.

## B.2 Construction of triplet training dataset

For image data, we first cluster the facial images based on identity similarity. From each cluster, we randomly select two images to form a target-source pair that shares the same identity but has different attributes. To generate the decoupling image, which has a different identity but the same attribute, we use the open-sourced InsightFace to create synthetic images with a distinct identity, while preserving the gender of the original face. This approach helps to avoid the degradation in quality observed when the original and swapped faces belong to different genders. Additionally, we exclude triplets with significant facial expression discrepancies by comparing the face landmarks. For video data, the process is similar, except that both the source and target images come from the same video clip, but not from the same frames as the target or motion images, which reduces the pose variation. Since video data is less abundant than image data, clustering does not yield enough pairs to form a sufficient number of triplets.

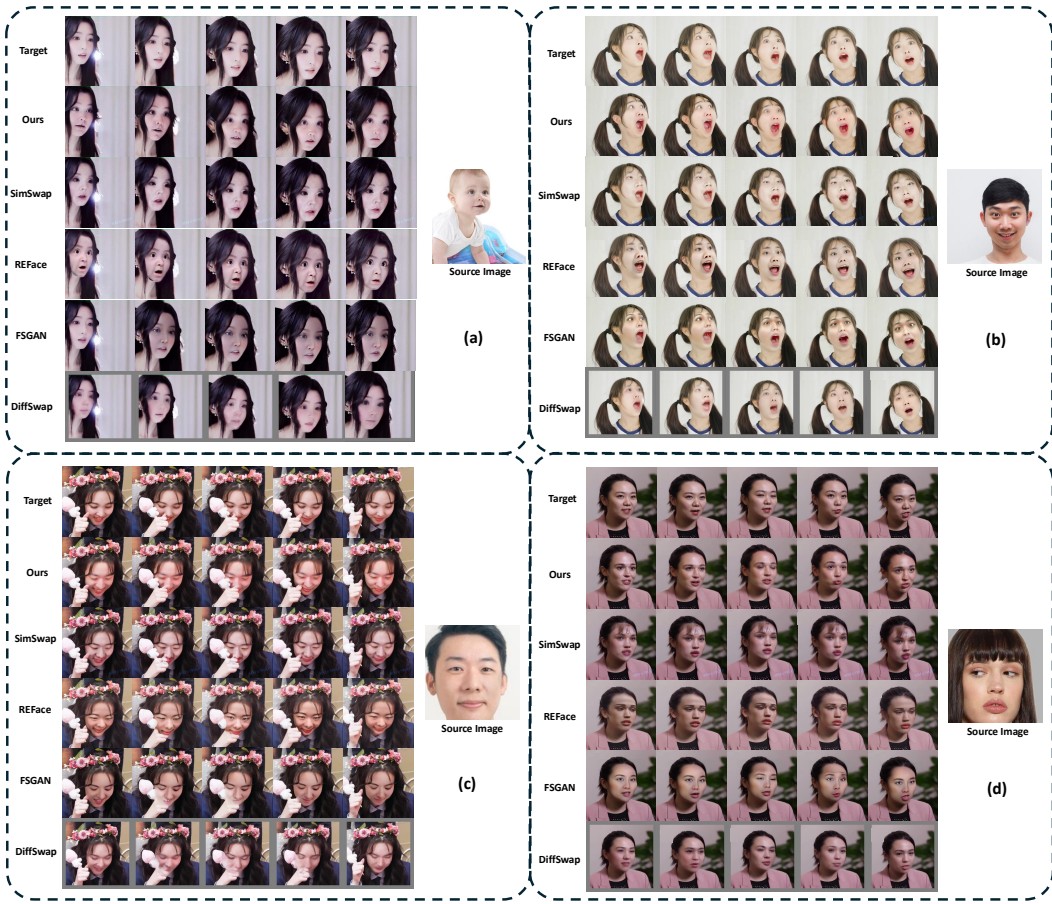

Figure 9: Qualitative comparison at $512 \times 512$ resolution. Our method generates high-fidelity results and handles challenging cases effectively, such as large poses (b) and occlusions (c). Corresponding videos are provided in the supplementary material.

## C More Visualization

In Figure 9, we show additional frames from the same video presented in the main text (Figure 6). We further provide supplementary visual examples in Figure 10, Figure 11, Figure 12, Figure 13, Figure 14, and Figure 15. The corresponding video results for all examples are included in the supplementary materials.

## D Limitations

While VividFace demonstrates strong temporal consistency, high-fidelity visual quality, and robustness in challenging scenarios, it also inherits certain limitations associated with diffusion-based architectures. Specifically, compared to GAN-based or lightweight encoder-decoder approaches, diffusion-based models typically require more inference steps to generate each frame, resulting in slower runtime performance. This can limit the practical applicability of our method in real-time or latency-sensitive scenarios, such as live video processing or interactive applications. Although we adopt efficient design choices and optimize the number of inference steps, the trade-off between generation speed and output quality remains an open challenge. Future work could explore accelerated diffusion sampling strategies or hybrid approaches that combine the strengths of both diffusion and feed-forward architectures to further improve efficiency without compromising quality.

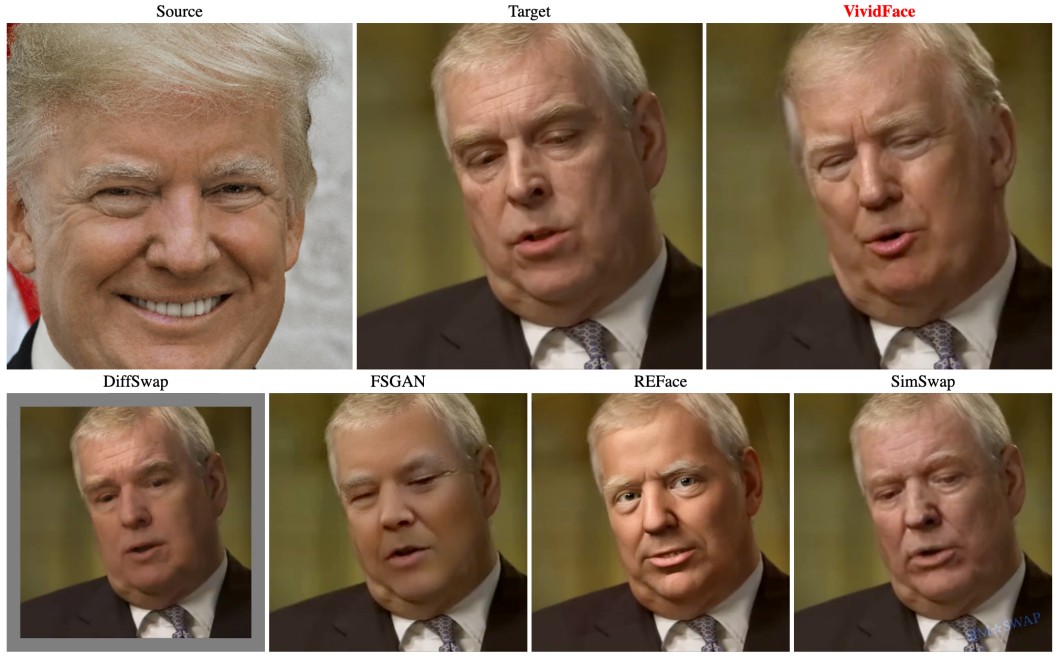

Figure 10: Qualitative comparison at $512 \times 512$ resolution. Corresponding videos are provided in the supplementary material.

# E Broader Impacts

VividFace advances video face swapping with high visual fidelity and temporal consistency, offering potential positive societal impacts in several domains. These include privacy protection through identity anonymization, safer film and content production by reducing the need for risky physical stunts, the creation of digital avatars in virtual environments, and accessible tools for individuals with communication or appearance-related challenges. However, the same capabilities may lead to negative societal consequences if misused. High-quality face-swapping techniques can be exploited for generating deepfakes, which may contribute to misinformation, identity theft, and other forms of digital deception. These risks are particularly relevant in political, social, and journalistic contexts where visual integrity is crucial. To address these concerns, we plan to release our models and datasets with strict usage terms and clear documentation. We encourage responsible use and support the development of detection tools and watermarking techniques to distinguish generated content from real footage. Furthermore, we advocate for continued dialogue in the research community around ethical deployment, regulatory considerations, and public education to minimize harm while enabling beneficial use cases. As with any powerful generative technology, the societal impact of VividFace will depend not only on the tool itself, but on how it is governed and integrated into real-world systems.

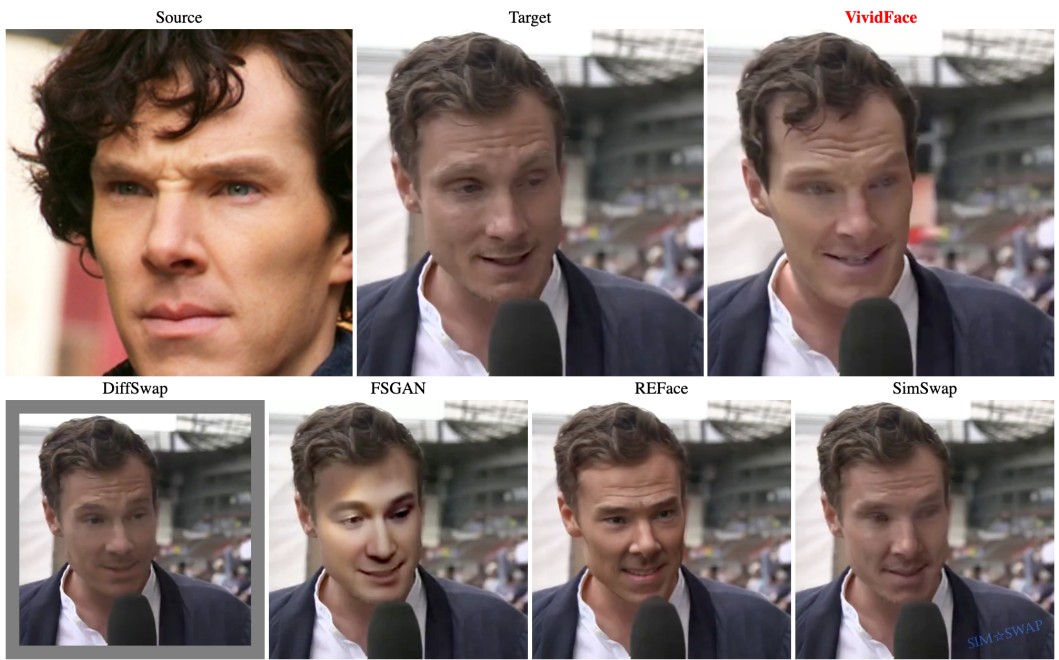

Figure 11: Qualitative comparison at $512 \times 512$ resolution. Corresponding videos are provided in the supplementary material.

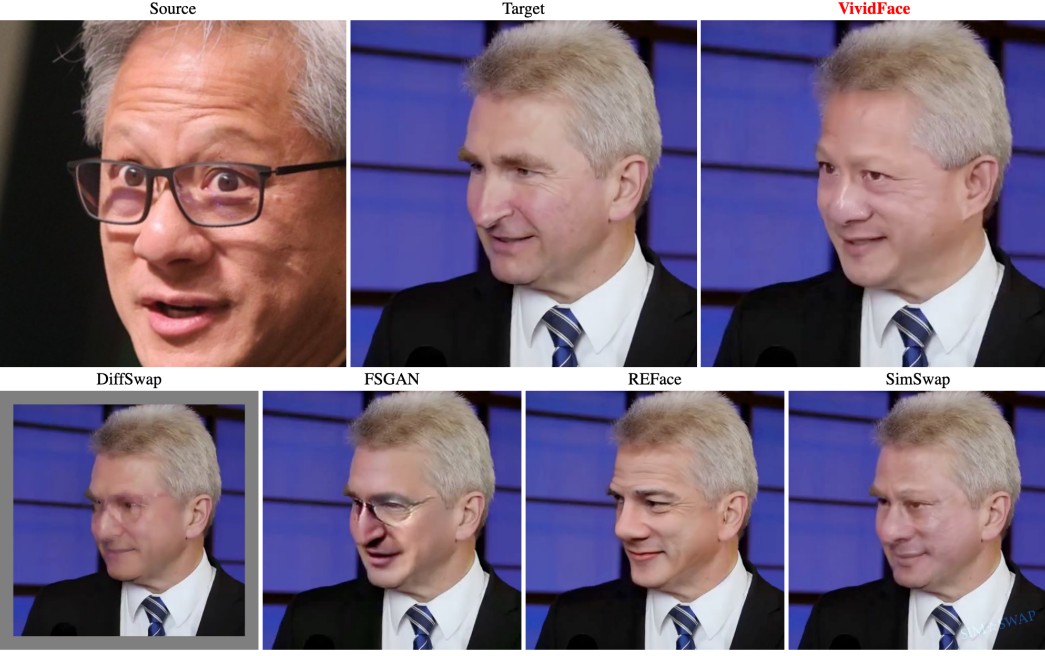

Figure 12: Qualitative comparison at $512 \times 512$ resolution. Corresponding videos are provided in the supplementary material.

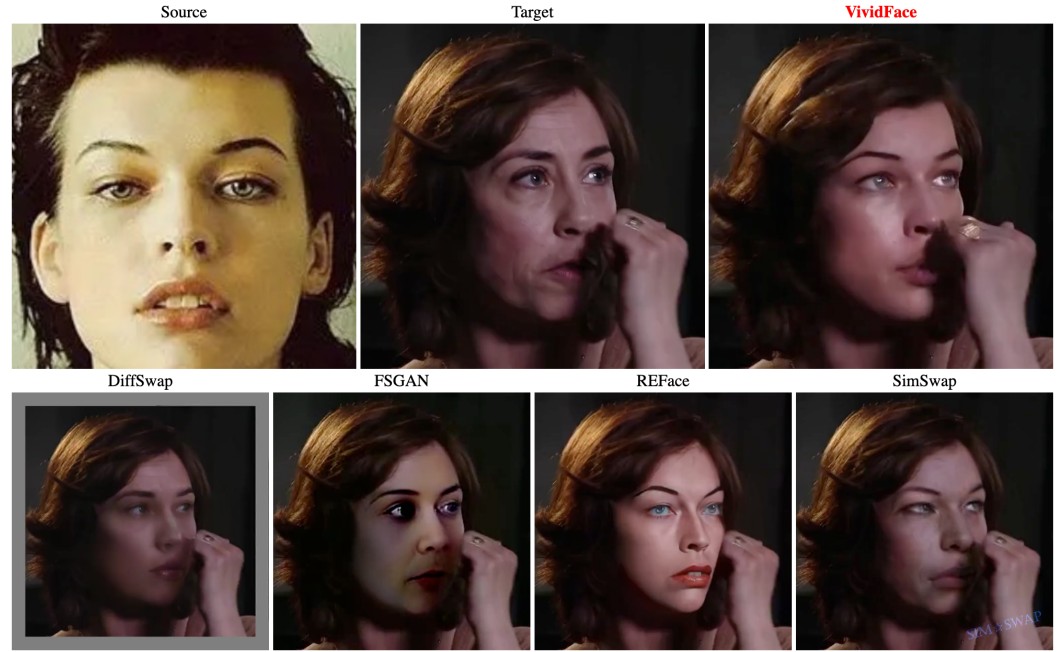

Figure 13: Qualitative comparison at $512 \times 512$ resolution. Corresponding videos are provided in the supplementary material.

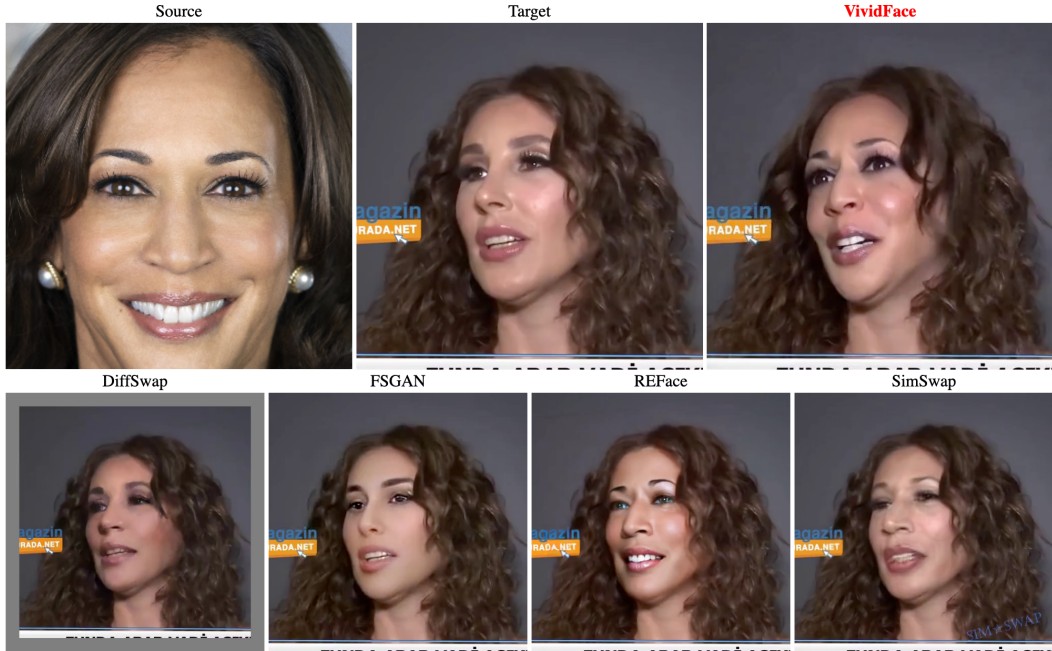

Figure 14: Qualitative comparison at $512 \times 512$ resolution. Corresponding videos are provided in the supplementary material.

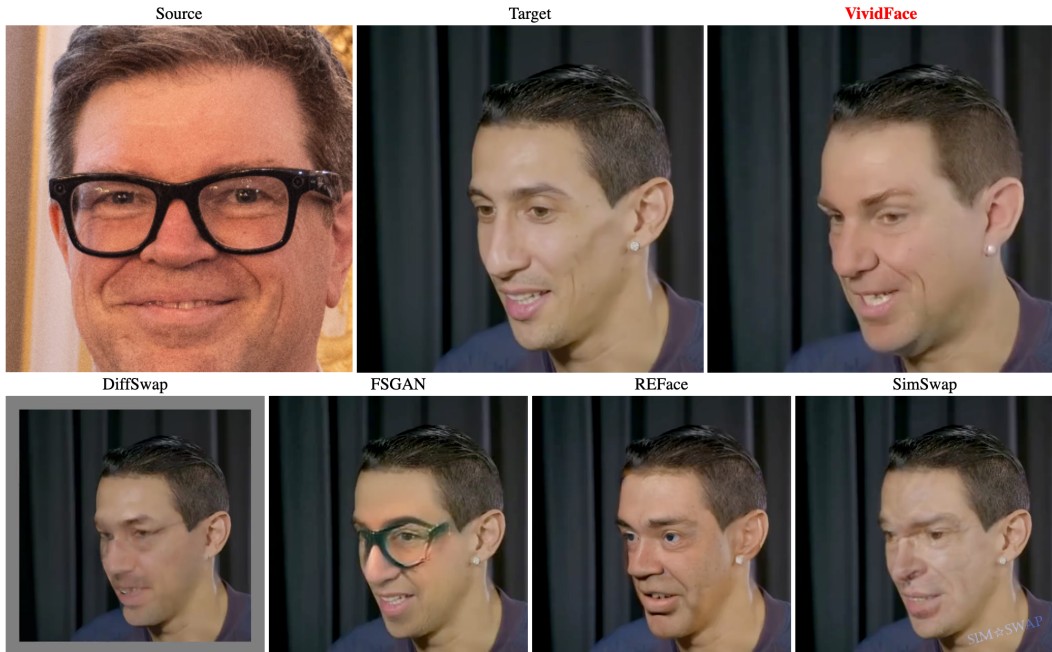

Figure 15: Qualitative comparison at $512 \times 512$ resolution. Corresponding videos are provided in the supplementary material.

