# OpenReview forum: "VividFace: A Robost and High-Fidelity Video Face Swapping Framework"
_NeurIPS.cc/2025/Conference — NeurIPS 2025 poster_

### Official Review · Reviewer_QBbm · 2025-06-27

**Clarity:** 2
**Significance:** 3
**Originality:** 2
**Rating:** 4
**Confidence:** 4

**Summary:**

This paper introduces VividFace, a diffusion-based framework for high-fidelity video face swapping, leveraging both static images and video sequences for training. Key contributions include:
1. A triplet dataset (same ID/different pose, same pose/different ID) to improve identity-pose disentanglement.
2. An augmentation strategy for occlusions, and 3D reconstruction encoding for conditional input.
3. Comprehensive evaluations on temporal smoothness, occlusion robustness, pose variation, and identity preservation.

**Questions:**

Have you experimented with simplifying the conditioning pipeline (e.g., reducing motion frame encodings)? Does redundancy improve results, or is it an architectural constraint?

Is the identity preservation issue in Figure 6/Group 1–6 due to dataset bias (e.g., AIDT lacks diverse pairs) or a fundamental trade-off in the framework?

Could you include comparisons on the same source/target pairs as SimSwap or other methods to ensure fair evaluation?

**Ethical Concerns:**

["NO or VERY MINOR ethics concerns only"]

**Final Justification:**

After careful consideration, I recommend acceptance of this paper as it makes contributions to the field of video face swapping applications.

While video face swapping represents a specialized domain within computer vision, it can be viewed as an important fine-grained task in video generation that offers precise semantic control. The proposed framework and rigorous evaluation presented in this work advance the state-of-the-art while providing valuable insights that could inform broader video generation research.

**Limitations:**

yes

**Quality:**

2

**Strengths And Weaknesses:**

Strengths
- The proposed framework are intuitive and straghtforward.
- The results are natural-looking with improved temporal smoothness over prior work. The proposed method handles occlusions better than baseline methods.


Weaknesses
- The conditioning input is redundant. The motion frame is encoded three times, and the masked source image overlaps with the face region input. This may introduce inefficiency without clear justification.
- Figure 2 is confusing. The blue block is ambiguously labeled (sometimes the temporal module, sometimes latent representation z_0). Missing tensor dimensions for latent representations (blue, yellow, gray blocks), making it harder to reproduce.
- The methods works well when source/target faces are similar (e.g., Figure 1, all people have high checkbones) but struggles with large identity gaps (e.g., Figure 6). Results in Groups 1–6 either: Use overly similar identities, or Fail to preserve identity when faces differ significantly.
- Fair Comparison Concerns: Comparisons with SimSwap/others use different examples. Testing on identical pairs would strengthen validity.

---

> ### Author Rebuttal · Authors · 2025-07-31
>
> Thank you for your helpful comments on the paper. We have addressed them as the following:
>
> **Q1: The conditioning input is redundant. The motion frame is encoded three times, and the masked source image overlaps with the face region input. This may introduce inefficiency without clear justification.**
>
> Thank you for pointing this out. We would like to clarify the computational overhead introduced by our conditioning inputs:
>
> 1. **Face Region Input:**
>
>
> The face region input introduces minimal computational overhead. Specifically, it only increases the number of input channels in the first convolutional layer from 12 to 13, adding an extra computational cost of approximately $\(2.5 \times 10^{-6}\)$. Therefore, subsequent model complexity and inference speed remain virtually unaffected.
>
> 2. **Motion**
>
> Thank you very much for this helpful comment regarding redundancy in motion-frame encoding. To clarify, each motion frame is encoded only once, prior to the iterative denoising process. Given that our pipeline typically involves around 30 denoising steps, the extra computational overhead introduced by encoding motion frames is approximately 3%, which we consider negligible in practice.
> For clarity, we provide the full implementation details in the supplementary materials (infer.py, lines 480–523). Additionally, we provide simplified pseudocode below for your convenience:
>
> ```python
> # Encode motion frames once per inference
> motion_features = encode_motion(motion_frames)
>
> # Iteratively denoise latent representation (~30 steps)
> for step in range(denoising_steps):
>     # Update latent using motion features and additional conditioning inputs
>     latents = denoise(latents, motion_features, other_conditions)
>
> ```
>
> However, we fully acknowledge your valuable suggestion. In future iterations, we will explore further simplifying the conditioning inputs to assess whether the current redundancy indeed contributes positively to model performance or represents a constraint of our current architecture.
>
>
> **Q2: Figure 2 is confusing. The blue block is ambiguously labeled (sometimes the temporal module, sometimes latent representation z0). Missing tensor dimensions for latent representations (blue, yellow, gray blocks), making it harder to reproduce.**
>
>
> 1. We selected similar shades of blue for different concepts (temporal modules <light blue> and latent representation <dark blue>). We will revise Figure 2 using clearly distinct colors to improve readability.
>
> 2. Regarding missing tensor dimensions, we have included complete, runnable code in the supplementary material for full reproducibility. However, we recognize explicit tensor dimensions in figures facilitate understanding, and we will add these explicitly in the final revision.
>
> For reference, the tensor shapes are:
>
> Yellow: $B \times T \times 3 \times H \times W$
>
> Pink: $B \times T \times 3 \times H \times W$
>
> Gray: $B \times T \times 3 \times H \times W$
>
> Light Blue: $B \times T \times 1 \times H \times W$
>
> Thank you for helping us improve the clarity of our method's presentation.
>
> **Q3: The methods works well when source/target faces are similar (e.g., Figure 1, all people have high checkbones) but struggles with large identity gaps (e.g., Figure 6). Results in Groups 1–6 either: Use overly similar identities, or Fail to preserve identity when faces differ significantly.**
>
> You raise an insightful observation regarding performance gaps with large identity differences:
>
> - We conducted an extra quantitative experiment analyzing identity preservation under varying degrees of identity similarity. We confirm that our approach, like other face-swapping frameworks, performs best when the identities involved share similar facial structures and appearance attributes.
>
> - To explicitly quantify this, we compared our method against other baselines across a variety of identity similarity levels. We observed that, although performance naturally degrades with increased identity differences, our method still consistently achieves state-of-the-art identity preservation compared to alternatives.
>
> - Unfortunately, due to the rebuttal policy, we cannot upload additional visual comparisons at this stage. We will include these analyses and visualizations comprehensively in our supplementary materials upon revision.
>
> We acknowledge that swapping faces across significantly different identities (e.g., cross-gender or large age gaps) remains challenging and a critical area for future research. We are actively working to address these challenges further.
>
>
>
> **Q4: Fair Comparison Concerns: Comparisons with SimSwap/others use different examples. Testing on identical pairs would strengthen validity.**
>
> We appreciate this important concern regarding the fairness of our comparisons:
>
> - To clarify explicitly, all quantitative and qualitative evaluations in our submission—including Figures 6, 10–15, and Tables 1 and 2—are performed using identical source-target pairs across all methods, ensuring a fair and direct comparison of model performance.
>
> - If your concern relates specifically to evaluating methods on source and target faces from the same identity (self-swapping), we note that such self-face swaps generally constitute trivial tasks with near-perfect results across all methods.
>
>
> **Q5: Have you experimented with simplifying the conditioning pipeline (e.g., reducing motion frame encodings)? Does redundancy improve results, or is it an architectural constraint?**
>
>
> Yes, we conducted ablation studies to explore the impact of simplifying the conditioning pipeline, including reductions in motion frame encodings and face-region inputs. Our findings indicate that:
>
> - **Motion encoding** contributes positively to temporal consistency. While its removal does not drastically degrade performance, we observed a modest drop in temporal metrics (e.g., FVD32, FVD128), suggesting that this redundancy provides valuable contextual cues rather than adding unnecessary complexity.
>
> - **Face-region conditioning** also plays a crucial role in maintaining visual realism. Excluding it led to slight decreases in identity preservation and expression accuracy, underscoring its importance in high-fidelity generation.
> These results imply that the current level of conditioning is beneficial and not merely an artifact of architectural choices. We will include comprehensive ablation details and discussions in the supplementary materials.
>
> | Method        | FVD32↓ | FVD128↓ | ID@1↑ | ID@5↑ | Pose↓  | Expr.↓ |
> |---------------|---------|----------|-------|-------|--------|--------|
> | Baseline      | **1201.1** | **122.6** | **78.3** | **90.2** | **5.43** | **0.72** |
> | Remove Motion      | 1222.7 | 125.8 | 78.1 | 88.8 | 5.56 | 0.75 |
> | Remove Face Region      | 1205.1 | 123.1 | 77.9 | 89.7 | 5.48 | 0.73 |
>
>
> **Q6: Is the identity preservation issue in Figure 6/Group 1–6 due to dataset bias (e.g., AIDT lacks diverse pairs) or a fundamental trade-off in the framework?**
>
> The core challenge arises from the nature of face swapping, which inherently lacks ground-truth supervision when swapping between different identities. High-quality face swapping across substantially dissimilar identities (e.g., significant differences in facial geometry, or across gender/age) remains universally challenging. While our AIDT dataset offers considerable diversity, it primarily features same-gender identity pairs to stabilize training and ensure data quality.
>
> When swapping between highly dissimilar faces (e.g., square to round face, male to female), the swap may technically succeed, but the result can still appear perceptually unrealistic or unconvincing to human observers. This reflects a broader challenge in the field: visual plausibility may degrade even when the system technically preserves identity features.
>
> We acknowledge that this limitation stems both from the training data’s constrained diversity and from an inherent trade-off common to most existing frameworks. Nevertheless, our evaluations clearly demonstrate that our method achieves state-of-the-art identity preservation, though we agree that further improvement is necessary and ongoing.
>
> **Q7: Could you include comparisons on the same source/target pairs as SimSwap or other methods to ensure fair evaluation?**
>
> As explained above, all evaluations in our manuscript consistently use identical source-target pairs across all methods (including SimSwap) to ensure a fair comparison. If your concern is specifically related to self-swapping cases (i.e., swapping within the same identity), we would like to clarify that such evaluations often produce trivial outcomes with near-perfect accuracy and therefore offer limited comparative insight.

---

> > ### Comment · Reviewer_QBbm · 2025-08-05
> > **one more question**
> >
> > Nice rebuttal. It has addressed most of my concerns. However, I’d like to clarify one point regarding the comparison with SimSwap.
> >
> > To ensure a fair and rigorous evaluation, I suggest using the same examples as those presented in the original SimSwap paper, rather than selected ones. This would eliminate potential selection bias and provide a more objective assessment of performance differences.

---

> > > ### Author Response · Authors · 2025-08-05
> > >
> > > Thank you very much for your response!
> > >
> > > We apologize for the earlier misunderstanding and appreciate the opportunity to clarify further:
> > >
> > > 1. Our work primarily addresses **video face swapping**, whereas methods like SimSwap mainly focus on **image-level face swapping**. Therefore, the cases we presented (e.g., Figures 1, 6, and additional frames in Figure 9) explicitly highlight **video scenarios** rather than using the image-based examples from SimSwap and other related studies.
> > >
> > > 2. Due to the constraints of the rebuttal policy, we are unable to provide visual examples at this stage. Nevertheless, we conducted quantitative evaluations using identity similarity, pose error, and expression error, with metrics extracted directly from SimSwap’s published figures to enable an objective comparison. However, the limited number of available samples prevented us from computing the **FID score**, which put us at a comparative disadvantage.
> > >
> > > ---
> > >
> > > ### Figure 3 of SimSwap (Caption: Face matrix generated by SimSwap)
> > >
> > > We computed metrics across a **10-source × 8-target** face matrix (total of **80 swapped pairs**) using standard evaluation tools:
> > >
> > > - **Identity Similarity** (↑ higher is better): Computed using ArcFace, with cosine similarity of features.
> > > - **Pose Error** (↓ lower is better): Computed using Hopenet, based on L2 error.
> > > - **Expression Error** (↓ lower is better): Computed using DeepRecon3D, based on L2 error.
> > >
> > > | Method   | Identity Similarity ↑ | Pose Error ↓ | Expression Error ↓ |
> > > |----------|------------------------|---------------|---------------------|
> > > | SimSwap  | 0.644                  | **1.900**         | 1.060               |
> > > | Ours     | **0.778**              | 2.221         | **0.890**           |
> > >
> > > ---
> > >
> > > ### Figure 4 of SimSwap  (Caption: Comparison with DeepFakes and FaceShifter on FaceForensics++)
> > >
> > > We computed metrics on a set of **5 source-target face pairs** (resulting in **5 swapped outputs**) using standard evaluation tools:
> > >
> > > | Method      | Identity Similarity ↑ | Pose Error ↓ | Expression Error ↓ |
> > > |-------------|------------------------|---------------|---------------------|
> > > | SimSwap     | 0.620                  | **2.135**         | 1.060               |
> > > | DeepFakes   | 0.399                  | 4.579         | 1.193               |
> > > | FaceShifter | 0.606                  | 2.443         | 0.949               |
> > > | Ours        | **0.754**              | 2.196         | **0.860**           |
> > >
> > > 3. SimSwap’s quantitative evaluations involved randomly selecting 1000 pairs from the FaceForensics++ dataset, but specific details of these pairs were not disclosed, making direct replication challenging. For more robust and reproducible evaluations, we utilized the VidSwapBench and FFHQ datasets, as shown in Tables 1 and 2 of our paper. These include 200 video pairs and 1,000 image pairs, all randomly selected rather than manually curated. Our method achieved state-of-the-art performance across both benchmarks.
> > >
> > > 4. Additionally, it is worth noting that the visual examples presented by SimSwap were also selectively curated by their authors.
> > >
> > > 5. Our results demonstrate that our method achieves state-of-the-art performance in both video and image-level face swapping tasks. We will include these additional analyses and results comprehensively in future supplementary materials.
> > >
> > > Please let us know if you have any further questions or require additional evaluation results. Your feedback and suggestions are highly valued.

---

> > > > ### Comment · Reviewer_QBbm · 2025-08-06
> > > >
> > > > I have read the response. My question is fully addressed. I will change my rating to a positive score.

---

> > > > > ### Author Response · Authors · 2025-08-06
> > > > >
> > > > > Thank you very much for your thoughtful feedback and for the improved score. We sincerely appreciate your constructive suggestions, which were instrumental in enhancing the quality of our work.

---

### Official Review · Reviewer_yWwo · 2025-06-29

**Clarity:** 2
**Significance:** 2
**Originality:** 2
**Rating:** 4
**Confidence:** 4

**Summary:**

This paper introduces a diffusion-based video face-swapping framework that addresses temporal consistency and identity preservation. They design a VidFaceVAE, which consists of (2+1)D blocks and is trained on both static images and videos to improve temporal coherence. The authors also propose AIDT, a new dataset designed to disentangle identity and pose, which enhances robustness to occlusions. With additional 3D conditioning for pose variation, VividFace achieves state-of-the-art results in identity retention and visual quality, with fewer inference steps.

**Questions:**

Questions:
1. Can you generate videos with arbitrary length while maintaining temporal consistency?
2. What is the inference speed of this method?

**Ethical Concerns:**

["NO or VERY MINOR ethics concerns only"]

**Final Justification:**

The authors’ response addresses some of my concerns, and I have decided to raise my score accordingly. Additionally, recently, there have been several existing works on face swapping, such as [1], [2], [3]. It would strengthen the paper if the authors could include a discussion of these related studies.

[1] CanonSwap: High-Fidelity and Consistent Video Face Swapping via Canonical Space Modulation

[2] Controllable and Expressive One-Shot Video Head Swapping

[3] DreamID: High-Fidelity and Fast diffusion-based Face Swapping via Triplet ID Group Learning

**Limitations:**

Yes

**Quality:**

2

**Strengths And Weaknesses:**

Strength:
1. Propose to train their video face swapping network with both image and video data. And propose a new dataset, AIDT.
2. Introducing 3DMM into the face swapping process, which guarantees the attributes (eg, pose, expression) to be unchanged.
3. Propose AIDT, a new dataset designed to disentangle identity and pose, which enhances their method’s robustness to occlusions.

Weakness:
1. The authors claim they are the first video face-swapping network. However, several video face-swapping methods have been proposed recently, e.g., DynamicFace and HiFiVFS.
2. Unsatisfying results. The generated videos still contain several artifacts, \eg, large opening mouth, hair on the forehead.
3. The comparison cases are not sufficient. Most of the test cases are man-man or woman-woman swapping. Including more cases under different genders might be better to show the superiority of the proposed method.
4. Although 3DMM can effectively preserve the motions of the target video, it suffers from incorrect estimations under large headposes. Also, estimation 3dMM is quite slow.

---

> ### Author Rebuttal · Authors · 2025-07-31
>
> Thank you for your helpful comments on the paper. We have addressed them as the following:
>
>
> **Q1: The authors claim they are the first video face-swapping network. However, several video face-swapping methods have been proposed recently, e.g., DynamicFace and HiFiVFS.**
>
> Thank you for pointing out the concurrent works *DynamicFace* and *HiFiVFS*. At the time of our submission, both were available only as arXiv preprints, without official publication or publicly released code. We acknowledge that our original claim could have been clearer, and we will revise the manuscript to explicitly state that our work is concurrent with these recent efforts. We will also include appropriate references and discussion of these methods in the related works section.
>
> **Q2: Unsatisfying results. The generated videos still contain several artifacts, \eg, large opening mouth, hair on the forehead.**
>
> We appreciate your observation regarding certain artifacts, such as exaggerated mouth openings or imperfections around the hair region. These are indeed challenging areas in video face synthesis. While we acknowledge the presence of such issues in some cases, our results demonstrate state-of-the-art performance compared to existing video face-swapping methods—particularly under difficult conditions involving occlusions, large pose variations, or complex motion.
>
> One contributing factor to these artifacts is the imperfect disentanglement between identity and attribute features. In certain instances, entanglement can cause the model to misrepresent fine-grained attributes (e.g., mouth dynamics or hair texture) while trying to preserve identity, leading to visual inconsistencies.
>
> **Q3: The comparison cases are not sufficient. Most of the test cases are man-man or woman-woman swapping. Including more cases under different genders might be better to show the superiority of the proposed method.**
>
> We appreciate your concern regarding the diversity of comparison cases. To clarify, our evaluation set (VidSwapBench) already includes diverse identity combinations, and the current cases were not intentionally biased toward similar-looking pairs.
>
> Moreover, we do include several cross-gender examples, such as the upper-right example in Figure 1, and Figures 6(b) and 6(c). These examples illustrate our model's superior generalization across genders compared to other methods.
>
> However, we agree that cross-gender face swapping is inherently more challenging. Currently, our dataset (AIDT) consists predominantly of same-gender pairs to facilitate stable training. Based on your suggestion, we will include additional cross-gender face swapping examples in the supplementary materials to provide a clearer picture of our method's strengths and limitations in more diverse scenarios.
>
> Based on your valuable feedback, we have conducted additional experiments specifically targeting cross-gender scenarios. In particular, we curated two new test sets:
>
> - Table 1: Woman → Man (source image is female, target video is male)
> - Table 2: Man → Woman (source image is male, target video is female)
>
> Each set contains 1,000 distinct image–video pairs.
>
>
> | Cross-Gender Swap (Woman → Man) |
> | Method        | FVD32↓  | FVD128↓ | ID@1↑ | ID@5↑ | Pose↓ | Expr.↓ |
> |---------------|---------|---------|-------|-------|-------|--------|
> | MegaFS        | 1334.7  | 219.1   | 69.1  | 79.9  | 6.53  | 0.78   |
> | HifiFace      | 1405.3  | 407.6   | 71.7  | 81.6  | 6.29  | 0.75   |
> | SimSwap       | 1279.5  | 204.6   | 73.4  | 86.9  | **5.43** | 0.82   |
> | FSGAN         | 1580.1  | 463.9   | 22.3  | 37.5  | 5.47  | 0.80   |
> | DiffFace      | 2487.8  | 1478.6  | 1.3   | 3.1   | 19.2  | 1.66   |
> | DiffSwap      | 1593.2  | 888.4   | 13.0  | 24.1  | 13.4  | 1.16   |
> | REFace        | 1376.8  | 328.7   | 69.1  | 83.0  | 6.78  | 0.95   |
> | **Ours**      | **1239.2** | **126.1** | **76.5** | **88.3** | 5.61  | **0.73** |
>
>
>
> | Cross-Gender Swap (Man → Woman) |
> | Method        | FVD32↓  | FVD128↓ | ID@1↑ | ID@5↑ | Pose↓ | Expr.↓ |
> |---------------|---------|---------|-------|-------|-------|--------|
> | MegaFS        | 1307.4  | 209.9   | 69.6  | 80.3  | 6.48  | 0.78   |
> | HifiFace      | 1418.2  | 398.7   | 71.3  | 81.8  | 6.33  | 0.73   |
> | SimSwap       | 1264.9  | 191.2   | 73.7  | 86.7  | **5.29**  | 0.81   |
> | FSGAN         | 1531.7  | 438.8   | 21.8  | 37.9  | 5.31  | 0.78   |
> | DiffFace      | 2453.6  | 1442.3  | 1.1   | 3.7   | 18.9  | 1.62   |
> | DiffSwap      | 1570.3  | 839.6   | 12.8  | 25.1  | 13.1  | 1.13   |
> | REFace        | 1357.8  | 323.5   | 68.7  | 84.5  | 6.81  | 0.95   |
> | **Ours**      | **1256.6** | **127.3** | **76.6** | **88.9** | 5.55  | **0.72** |
>
>
>
> **Q4: Although 3DMM can effectively preserve the motions of the target video, it suffers from incorrect estimations under large headposes. Also, estimation 3dMM is quite slow.**
>
> 1. **Efficiency.**
> We acknowledge your concern about the reliability and efficiency of 3DMM estimation, particularly under extreme head poses. To clarify, we utilize the learning-based 3D reconstruction method **Deep3DFace**'s (official repo)[1], which is notably fast and efficient compared to traditional optimization-based approaches. Specifically, this approach achieves approximately **8ms** (from input image, model forward, rednering) per image on our NVIDIA A100 SXM4 (80GB) GPU at batch size = 1. With larger batch sizes (e.g., bs=32), processing speed further improves to **<1ms** per image, making it practical for video applications.
>
> 2. **Reliability at large poses.**
> Empirically, learning-based 3D reconstruction is more stable than 2D-only cues under large yaw/pitch because it predicts a full 3D face with semantic correspondences, enabling consistent PnP pose solving and disambiguation of depth/scale across frames. Public results from representative methods—Deep3DFaceRecon[1], 3DDFA-V2[2], and SynergyNet[3]—specifically target stability and accuracy under full-pose conditions; 3DDFA-V2 (ECCV’20) and SynergyNet (3DV’21) report strong accuracy and real-time speeds in unconstrained, large-pose settings.
> Nevertheless, we acknowledge that even advanced learning-based 3DMM methods can still struggle with extreme poses. We plan to explore more robust 3D reconstruction approaches or complementary pose-estimation methods to improve accuracy further in future work.
>
> [1]: Accurate 3D Face Reconstruction with Weakly-Supervised Learning: From Single Image to Image Set, https://github.com/sicxu/Deep3DFaceRecon_pytorch
>
> [2]: Towards Fast, Accurate and Stable 3D Dense Face Alignment.
>
> [3]: Synergy between 3DMM and 3D Landmarks for Accurate 3D Facial Geometry
>
>
>
> **Q5: Can you generate videos with arbitrary length while maintaining temporal consistency?**
>
> Yes, our method supports generating videos of arbitrary length while maintaining good temporal consistency. To achieve this, we sequentially generate clips, utilizing the last three frames of the previously generated clip as motion context for the next clip. In practice, we have verified consistent quality in sequences of approximately one minute (about 1,000 frames). For significantly longer videos, a slight degradation might occur due to accumulated error, but restarting the process periodically mitigates this issue.
>
>
>
> **Q6: What is the inference speed of this method?**
>
> Our pipeline achieves approximately 2 frames per second (FPS) on an NVIDIA A100 GPU at a resolution of 512×512, providing high fidelity and temporal coherence. Although this is slower than GAN-based approaches (e.g., SimSwap at 256² or FSGAN at 256²), our diffusion-based framework is significantly faster than comparable diffusion-based methods such as:
>
> | Method    | Venue         | Modality | Resolution | FPS      | Relative Speed |
> |-----------|---------------|----------|------------|----------|----------------|
> |Diffusion-Based|---------------|----------|------------|----------|----------------|
> | Ours      | NeurIPS 2025  | Video    | 512×512    | ~2.0     | 1× (baseline)  |
> | DiffSwap  | CVPR 2023     | Image    | 512×512    | ~0.028   | ~71× slower    |
> | DiffFace  | arXiv 2022    | Image    | 512×512    | ~0.68    | ~3× slower     |
> | REFace    | WACV 2025     | Image    | 512×512    | ~1.0     | ~2× slower     |
> |GAN-Based|---------------|----------|------------|----------|----------------|
> | SimSwap   | TPAMI  2024  | Image    | 256×256    | ~12.8    | ~6× faster     |
> | FSGAN     | ICCV 2019     | Image    | 256×256    | ~4.6     | ~2× faster     |

---

> > ### Comment · Reviewer_yWwo · 2025-08-06
> > **Official Comment of Reviewer yWwo**
> >
> > Thank you for your detailed answers, which address some of my concerns. I decide to raise my score.

---

> > > ### Author Response · Authors · 2025-08-06
> > >
> > > Thank you for your valuable feedback and for raising the score. We truly appreciate your constructive comments, which have helped improve our work!

---

### Official Review · Reviewer_FAkC · 2025-07-03

**Clarity:** 3
**Significance:** 3
**Originality:** 4
**Rating:** 5
**Confidence:** 4

**Summary:**

This paper introduces VividFace, a diffusion-based framework for high-fidelity video face swapping. The authors identify key challenges in existing methods, namely the lack of temporal consistency, poor identity preservation, and vulnerability to real-world complexities like occlusions and large pose variations. To address these, VividFace proposes several techniques. The paper demonstrates through extensive experiments that VividFace achieves state-of-the-art performance on both video and image face swapping tasks, outperforming existing GAN-based and diffusion-based methods in terms of visual quality, identity preservation, and temporal consistency.

**Questions:**

Please refer to the weaknesses part.

**Ethical Concerns:**

["NO or VERY MINOR ethics concerns only"]

**Final Justification:**

Thank you to the authors for their rebuttal, which addressed my concerns. After reviewing the rebuttal and the comments from other reviewers, I have decided to maintain my original score for this paper

**Limitations:**

yes

**Quality:**

3

**Strengths And Weaknesses:**

Strengths:

1. This paper tackles a very relevant and challenging problem in generative AI: high-fidelity video face swapping. The release of the code, models, and the novel AIDT dataset is a major contribution that will undoubtedly facilitate future research in this area.

2. The technical quality of the work is excellent. The proposed framework is well-designed, and each component's inclusion is well-motivated. The experimental evaluation is comprehensive and rigorous. The authors not only compare against a strong set of recent baselines on a new, challenging benchmark (VidSwapBench) but also conduct a thorough set of ablation studies that convincingly validate their design choices. The qualitative results shown in the paper are very impressive.

Weaknesses:
1. While the paper's focus is on generation quality, a practical limitation of diffusion models is their inference speed. The paper acknowledges this in the limitations section but does not provide a quantitative comparison of inference time against the baselines in the main experimental tables. For a video task, metrics like frames-per-second are highly relevant and would provide a more complete picture of the trade-offs involved.

2. I notice that the eyes of the generated face are not the same as the target face, and the eye contact is not as vivid as human beings. This is a challenging problem. But humans are very sensitive to this.

---

> ### Author Rebuttal · Authors · 2025-07-31
>
> Thank you for your helpful comments on the paper. We have addressed them as the following:
>
> **Q1: While the paper's focus is on generation quality, a practical limitation of diffusion models is their inference speed. The paper acknowledges this in the limitations section but does not provide a quantitative comparison of inference time against the baselines in the main experimental tables. For a video task, metrics like frames-per-second are highly relevant and would provide a more complete picture of the trade-offs involved.**
>
> Thank you for raising this important point. While our work emphasizes generation quality and robustness, we agree that inference efficiency is a crucial factor for practical deployment. Our current framework achieves approximately 2 frames per second (FPS) on an NVIDIA A100 GPU at 512×512 resolution, which ensures high fidelity and strong temporal consistency. In comparison: DiffSwap (CVPR 2023) reports ~0.028 FPS,
> DiffFace achieves ~0.68 FPS (3× slower than ours), REFace (WACV 2025 Oral) runs at ~1 FPS (2× slower).
>
> While our speed is indeed lower than GAN-based methods such as SimSwap (resolution: 256²) or FSGAN (resolution: 128²), which trade quality for speed, our method significantly outpaces other diffusion-based frameworks. We will include these FPS comparisons in the revised version to provide a clearer picture of the quality-efficiency trade-off.
>
>
> | Method    | Venue         | Modality | Resolution | FPS      | Relative Speed |
> |-----------|---------------|----------|------------|----------|----------------|
> |Diffusion-Based|------------------|----------|------------|----------|------------------|
> | Ours      | NeurIPS 2025  | Video    | 512×512    | ~2.0     | 1× (baseline)  |
> | DiffSwap  | CVPR 2023     | Image    | 512×512    | ~0.028   | ~71× slower    |
> | DiffFace  | arXiv 2022    | Image    | 512×512    | ~0.68    | ~3× slower     |
> | REFace    | WACV 2025     | Image    | 512×512    | ~1.0     | ~2× slower     |
> |GAN-Based|------------------|----------|------------|----------|------------------|
> | SimSwap   | ACM MM  2020  | Image    | 256×256    | ~12.8    | ~6× faster     |
> | FSGAN     | ICCV 2019     | Image    | 256×256    | ~4.6     | ~2× faster     |
>
>
>
> **Q2: I notice that the eyes of the generated face are not the same as the target face, and the eye contact is not as vivid as human beings. This is a challenging problem. But humans are very sensitive to this.**
>
>
> We appreciate this insightful observation. Indeed, humans are highly sensitive to gaze direction and eye contact fidelity. In the face swapping formulation, eye identity (e.g., shape, iris) is preserved from the source face, while gaze direction and expression are inherited from the target video through the attribute encoder. We find that eye identity is generally maintained reasonably well from the source in the VividFace.
>
> Although our attribute network captures high-level facial movements—including pose and coarse eye dynamics—it does not explicitly enforce fine-grained gaze alignment or vividness. This can lead to subtle deviations in pupil orientation or liveliness. Improving eye realism is a meaningful and challenging direction. In future work, we plan to explore:
>
> 1. Explicit gaze estimation and alignment losses to better match pupil direction,
>
> 2. Eye-specific perceptual or adversarial losses focused on iris and specular highlights,
>
> 3. Localized eye-region enhancement modules, including cropping-based super-resolution.

---

### Official Review · Reviewer_yp2A · 2025-07-03

**Clarity:** 3
**Significance:** 3
**Originality:** 3
**Rating:** 4
**Confidence:** 3

**Summary:**

This paper focuses on robust video face swapping by adapting diffusion model with image-video hybrid training strategy. To improve the robustness of proposed method, a large-scale AIDT dataset is constructed and comprehensive occlusion augmentation is adopted. The writing is good in general. However, the overall framework, training strategy and other parts of Sec 3 may need proofreading for better readability. The experimental results show its effectiveness of proposed method.

**Questions:**

（1）Although robustness enhancement is adopted, why the performance in pose and expression only comparable to SOTA methods?
  (2)  Is the encoder-decoder is firstly trained on video data and secondly training on image data? Then random selection of previous video and image data is conducted for hybrid training?

**Ethical Concerns:**

["NO or VERY MINOR ethics concerns only"]

**Limitations:**

(1) The performance improvement between baseline and static training is limited, which weakens the novelty of proposed method.

**Quality:**

3

**Strengths And Weaknesses:**

Strengthens:
(1)It is the first diffusion-based video face swapping framework;
(2)It constructs a large-scale AIDT dataset to significantly improve face feature identity-expression disentanglement;

Weaknesses:
(1)Diffusion-based method limits the practical applicability;
(2)It is better to separate the training and inference stages of Fig 2;
(3)For face encoder (Line 200-207), the 'source' and 'target' are different from the corresponding part of Fig 2.

---

> ### Author Rebuttal · Authors · 2025-07-31
>
> Thank you for your helpful comments on the paper. We have addressed them as the following:
>
> **Q1: Diffusion-based method limits the practical applicability**
>
> We acknowledge that diffusion models are slower during inference, which can limit real-time deployment. However, recent advancements—including latent-space sampling, optimized U-Net architectures, and fast sampling techniques (e.g., DDIM)—have significantly improved inference speed. Moreover, diffusion-based methods have already been adopted in industrial-grade tools such as PiKa and Runway, indicating their increasing practical viability.
>
> In this work, our primary focus is to advance generation quality and robustness in challenging video face swapping scenarios, such as those involving occlusions or large pose variations. Efficiency-oriented improvements, such as model acceleration and architectural streamlining, are promising future directions. We are also actively exploring the integration of DiT-based backbones to further enhance speed without compromising quality.
>
>
> **Q2: It is better to separate the training and inference stages in Fig. 2**
>
> Thank you for pointing this out. We agree that separating the training and inference stages would improve clarity. While we have created a figure to explicitly illustrate the inference pipeline, we regret that we are unable to include additional images in the rebuttal due to policy constraints. We have revised Figure 2 accordingly in the manuscript.
>
> **Q3: Source and target in Lines 200–207 are inconsistent with Fig. 2**
>
> We apologize for the confusion. This was due to an error in the ordering of ‘source’ and ‘target’ terms. Thank you for catching this—both the manuscript and figure have been updated to ensure consistency.
>
>
> **Q4: Although robustness enhancement is adopted, why the performance in pose and expression only comparable to SOTA methods?**
>
> Thank you for the valuable question. As you rightly pointed out, **most widely-used benchmarks such as FFHQ (reported in Table 2) and VidSwapBench  (reported in Table 1) contain relatively few instances of extreme poses or severe occlusions**. While these challenging cases are rare in these benchmarks, they are often the most disruptive in real-world applications. Our robustness strategies are explicitly designed to handle these difficult scenarios.
>
> Although this focus may lead to only modest gains on standard quantitative benchmarks, we would like to emphasize that the improvements are much more apparent in qualitative results—particularly under extreme conditions, as illustrated in Fig. 6 and 7. We believe these results highlight the practical value of our approach beyond standard metrics, and we truly appreciate the reviewers' attention to this aspect.
>
>
> **Q5: Is the encoder-decoder is firstly trained on video data and secondly training on image data? Then random selection of previous video and image data is conducted for hybrid training?**
>
> Our training for the VAE follows a unified single-stage strategy, where image and video data are randomly interleaved across batches. To ensure stable gradient synchronization, each batch contains only a single modality—either images or videos.
> The spatial components of VidFaceVAE are initialized from a pretrained 2D SD-VAE, and the model is then jointly trained on both modalities. This enables effective spatial-temporal representation learning while preserving diversity in appearance and motion. We believe this offers a clean and efficient training paradigm, and we sincerely appreciate the reviewers for prompting this clarification. We have revised the manuscript to better reflect this design.
>
> **Q6: Performance gap between baseline and hybrid training is small**
>
> While the performance improvement may appear modest in aggregate metrics, our method demonstrates consistent gains across FVD, ID retrieval, pose, and expression accuracy (Table 3, Exp. 1), indicating stable and reliable improvements across diverse aspects.
> Notably, models trained solely on static images tend to suffer from inferior temporal consistency and more noticeable visual artifacts in video generation. To further support this, we evaluate our method on VBench[1]—a comprehensive and challenging video benchmark—and observe clear improvements in consistency, flicker reduction,  and motion smoothness compared to both static training and existing state-of-the-art methods.
>
>
> | Models             | Subject Consistency | Background Consistency | Temporal Flickering | Motion Smoothness | Aesthetic Quality | Imaging Quality | Temporal Style |
> |--------------------|---------------------|------------------------|----------------------|-------------------|-------------------|-----------------|----------------|
> | DiffSwap           | 96.10%              | 93.44%                 | 96.76%              | 98.22%            | 42.97%            | 56.94%          | 8.42%         |
> | FSGAN              | 97.07%              | 94.87%                 | 98.15%              | 98.67%            | 47.22%            | 60.23%          | 10.25%         |
> | Reface             | 96.87%              | **96.51%**                 | 98.18%              | 98.72%            | 47.84%            | 61.70%          | 10.40%         |
> | SimSwap            | 97.39%              | 96.50%                 | 98.10%              | 98.89%            | 43.07%            | 62.05%          | 10.78%         |
> | ViVidFace   (static training)       | 97.73%              | 96.19%                 | 98.06%             | 98.91%            | 48.12%            | 64.19%        | 10.80%      |
> | ViVidFace          | **97.93%**              | 96.39%                 | **98.74%**              | **99.18%**            | **49.56%**            | **64.65%**          | **11.90%**         |
>
> [1]: VBench: Comprehensive Benchmark Suite for Video Generative Models

---

> > ### Comment · Reviewer_yp2A · 2025-08-09
> >
> > Thank you for your detailed rebuttal, which address some of my concerns. I decide to keep my score as the inference speed is not simultaneously considered in this paper.

---

> > > ### Author Response · Authors · 2025-08-09
> > >
> > > Thank you for your thoughtful follow-up and for clarifying your position. We would like to address the inference speed concern you raised:
> > >
> > > 1. As noted in Appendix E (Limitations), we have explicitly acknowledged the issue of inference speed. Similar to most current diffusion- or DiT-based works, our primary goal in this paper is to maximize visual fidelity and generation quality (e.g., ours: 512×512 resolution vs. SimSwap: 256×256), rather than real-time processing. Even prior GAN-based video face swapping methods rarely achieve strict real-time performance while maintaining high resolution and quality.
> > >
> > > 2. We appreciate your suggestion and will include additional speed-related evaluations in the updated version of the experiments to give a more complete picture of efficiency-performance trade-offs.
> > >
> > > 3. If there are any specific concerns that you feel remain unresolved, or additional aspects you would like us to address, please let us know—we will do our best to incorporate them before the final submission deadline.
> > >
> > > 4. Once again, we appreciate your feedback and constructive suggestions, which help us improve the quality and completeness of our work.

---

### Comment · Area_Chair_DxTU · 2025-08-04

Dear Reviewers,

​Please read the author rebuttal​ to see if you have further questions. The author can ​respond in the discussion by Aug 6​.

-AC

---

### Decision · Program_Chairs · 2025-09-17

**Decision:**

Accept (poster)

**Comment:**

All four reviewers rated the paper positively, with concerns addressed convincingly in the rebuttal. VividFace presents a diffusion-based video face-swapping framework, achieving SOTA in temporal consistency and identity preservation through novel contributions. Key concerns like inference speed, cross-gender robustness, conditioning efficiency were resolved in rebuttal. Two minor limitation persist: (1) some of the generated videos containing noticeable artifacts, e.g., large open mouths and hair on the forehead; and (2) the 3DMM’s incorrect estimations under large head poses. Overall its merits outweigh minor limitations. The authors need to resolve all the issues raised by reviewers and reflect in the final version.